# Coverage-dependent bias creates the appearance of binary splicing in single cells

Carlos F Buen Abad Najar[1], Nir Yosef[1,2,3,4]*, Liana F Lareau[1,5]*

[1]Center for Computational Biology, University of California, Berkeley, Berkeley, United States; [2]Department of Electrical Engineering and Computer Science and the Center for Computational Biology, University of California, Berkeley, Berkeley, United States; [3]Ragon Institute of MGH, MIT, and Harvard, Cambridge, United States; [4]Chan Zuckerberg Biohub, San Francisco, San Francisco, United States; [5]Department of Bioengineering, University of California, Berkeley, Berkeley, United States

**Abstract** Single-cell RNA sequencing provides powerful insight into the factors that determine each cell's unique identity. Previous studies led to the surprising observation that alternative splicing among single cells is highly variable and follows a bimodal pattern: a given cell consistently produces either one or the other isoform for a particular splicing choice, with few cells producing both isoforms. Here, we show that this pattern arises almost entirely from technical limitations. We analyze alternative splicing in human and mouse single-cell RNA-seq datasets, and model them with a probabilistic simulator. Our simulations show that low gene expression and low capture efficiency distort the observed distribution of isoforms. This gives the appearance of binary splicing outcomes, even when the underlying reality is consistent with more than one isoform per cell. We show that accounting for the true amount of information recovered can produce biologically meaningful measurements of splicing in single cells.

*For correspondence:
niryosef@berkeley.edu (NY);
lareau@berkeley.edu (LFL)

Competing interests: The authors declare that no competing interests exist.

## Introduction

Single-cell RNA sequencing (scRNA-seq) has provided impressive temporal resolution to our understanding of continuous biological processes such as cell differentiation (*Wagner et al., 2016*; *Tanay and Regev, 2017*). It has uncovered hidden heterogeneity among cells and exposed the factors that determine each cell's unique identity. One broad source of transcriptomic diversity is alternative splicing, and several studies have uncovered compelling evidence of changes in alternative splicing among single cells during differentiation (*Welch et al., 2016*; *Qiu et al., 2017*; *Song et al., 2017*; *Huang and Sanguinetti, 2017*).

A particularly surprising conclusion of several scRNA-seq studies was the observation that splicing was often bimodal among supposedly homogeneous cells (*Shalek et al., 2013*; *Marinov et al., 2014*; *Song et al., 2017*; *Westoby et al., 2018*). Individual cells had binary outcomes in splicing: some cells always spliced in a particular cassette exon, and some cells never spliced in the exon, but few cells showed truly intermediate inclusion within one cell. This unexpected result contrasted with previous single molecule imaging studies of several alternative exons that showed that cell-to-cell variability is minimized and tightly regulated by the splicing machinery in single cells (*Waks et al., 2011*; *Maamar et al., 2013*). Curiosity about this result led to investigations of mechanisms that might be responsible for stochastic splicing variability among apparently homogeneous cells, such as variation in DNA methylation (*Linker et al., 2019*).

We propose that the observed bimodality does not generally reflect a binary nature of splicing biology, but rather, that it exposes a technical limitation of the scRNA-seq data that have been collected so far. Because alternative isoforms of a gene share much of the same sequence, only the few RNA-seq reads mapping to the exact alternative splice junctions, or to the alternative exon itself, reveal its alternative splicing. When combined with the low mRNA capture efficiency of scRNA-seq and the PCR amplification of small amounts of starting material into a full-length sequencing library, these circumstances create the risk of bottlenecks that lose all but a few individual mRNAs of most genes in each cell.

The limitations of scRNA-seq are a known obstacle to studying splicing in single cells (*Arzalluz-Luque and Conesa, 2018*). Similar concerns have arisen with the use of scRNA-seq to infer allelic expression; a careful analysis showed that stochastic patterns resulted from technical noise (*Kim et al., 2015*). A recent study observed and modeled the high dropout rate of individual isoforms in scRNA-seq and advised that scRNA-seq is fundamentally unsuitable for measuring changes in alternative splicing (*Westoby et al., 2020*). Others have implemented workarounds, for example using sequence features to predict splicing outcomes in lieu of sufficient sequencing coverage (*Huang and Sanguinetti, 2017*), or attempting to identify excess variance beyond technical noise (*Welch et al., 2016*; *Linker et al., 2019*). These studies have identified true examples of differential splicing in single cells, but they fundamentally do not explain how scRNA-seq limitations have caused qualitative, not just quantitative, distortions in our understanding of alternative splicing.

Here, we show that scRNA-seq splicing data are consistent with a simple model (*Figure 1*). Consider a particular cassette exon whose true pattern of exclusion follows a unimodal distribution of isoform ratios across cells (i.e. most cells express both isoforms, with a ratio revolving around the same mean). This distribution can be distorted by extreme information loss during library preparation and sequencing, creating the illusion that individual cells only produce one isoform or the other. Our simulations make it clear that the reliability of splicing measurements is a function of the initial amount of mRNA, the efficiency of its recovery, the underlying splicing rate, and further distortions from PCR amplification of cDNA. These effects should be considered when interpreting previous studies that used qualitative changes in the observed distribution of the splicing rates (*Song et al., 2017*) or changes in their variance (*Linker et al., 2019*) as evidence for regulation of alternative splicing. Considering the true amount of information available for a cassette exon can allow for accurate observations of alternative splicing. Using a data normalization and filtering method to identify cassette exons with sufficient information, we are able to draw biologically relevant conclusions about alternative splicing in single cells.

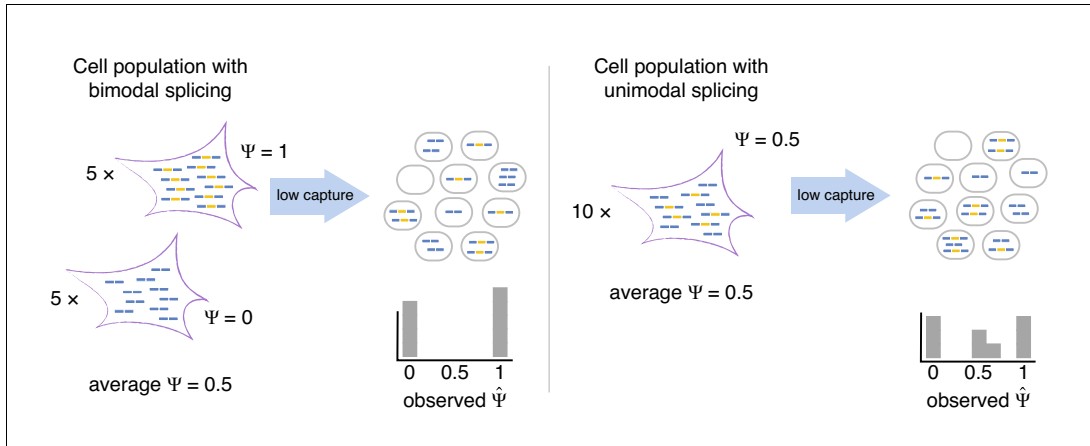

**Figure 1.** Bimodal vs unimodal models of cassette exon splicing. In the bimodal model, some cells consistently splice in the exon, while others consistently skip it. After mRNA capture and sequencing, observations of Ψ are almost exclusively binary. In the unimodal model, individual cells express some mRNAs that splice in the cassette exon and some that skip it. Low mRNA capture dramatically reduces the number of cells in which both isoforms are observed, artificially inflating binary Ψ values.

## Results

Our interest in splicing regulation led us to examine alternative splicing in several single cell differentiation datasets from mice and humans that were generated with methods that recover sequence from along the full length of mRNAs. To investigate the reported high variability of splicing between cells more closely, we began by examining the splicing of cassette exons in a high-coverage mouse scRNA-seq dataset (*Chen et al., 2016*), estimating their percent spliced-in as the fraction of splice junction reads that show exon inclusion (out of all reads that cover the junction). We use $\hat{\Psi}$ to denote these estimated rates, while $\Psi$ denotes the actual rate as it is in the cell. For clarity, we define a single $\hat{\Psi}$ observation (which pertains to a specific cassette exon in an individual cell) as *binary* if it is close to 0 or 1 (i.e. the respective cell tends to express transcripts that either include the exon or exclude it, but not both). We then describe the distribution of an exon's $\hat{\Psi}$ across cells as *bimodal* when its individual values are predominantly binary, where some cells have a $\hat{\Psi}$ close to 1 (most observed transcripts include the exon) and others have $\hat{\Psi}$ close to 0 (most observed transcripts do not include the exon). Strikingly, when we inspected several exons, we saw that they had more binary outcomes in cells with fewer reads covering their splice junctions, while cells with more reads were more likely to show non-binary $\hat{\Psi}$ values (*Figure 2a*). We realized that this effect of coverage may reflect a non-binary reality, since even if both isoforms are expressed in a certain cell, the likelihood of observing both isoforms is reduced as the number of captured mRNAs decreases. In contrast, if the underlying distribution were indeed bimodal with binary modes, as previously proposed (*Shalek et al., 2013*; *Marinov et al., 2014*; *Song et al., 2017*), then the read coverage would have little effect on the proportion of binary $\hat{\Psi}$ observations across cells.

To further explore this phenomenon, we extended our analysis to the full scRNA-seq datasets. In all cases, we found a strong effect of coverage on the observed binary $\hat{\Psi}$ in exons with intermediate splicing (average $\hat{\Psi}$ between 0.2 and 0.8). We consistently found that exons with low junction read coverage had more binary $\hat{\Psi}$ values and bimodal $\hat{\Psi}$ distributions (*Figure 2b,c*; *Figure 2—figure supplement 1a,b*). We found that the association between binary values and read coverage was not observed in exons that are binary but not bimodal (i.e. nearly constitutively excluded or included exons, with average $\hat{\Psi}$ close to 0 or 1; *Figure 2c*). Taken together, these observations suggest that the abundance of binary observations in exon inclusion patterns may reflect a distortion of an underlying unimodal splicing distribution (i.e. when cells in fact express both isoforms), rather than a truly bimodal splicing pattern in the analyzed cells.

We then asked if the association of binary splicing outcomes with low read coverage could be due to some biological consequence of low transcription, or if it was a technical consequence of low sequencing coverage. We found that, among the cells in a single experiment, the cells with an overall higher number of splice junction reads also tended to have a smaller fraction of exons with binary values (*Figure 2d*, *Figure 2—figure supplement 1b*), suggesting an influence of technical coverage rather than transcription level. We further considered the effect of biological variations in transcription. Transcription at a single locus occurs in intermittent bursts of RNA synthesis (*Raj and van Oudenaarden, 2008*), suggesting a possible effect of size and frequency of transcriptional bursts on binary splicing. Using transcriptional bursting data from *Larsson et al., 2019*, we found that genes with either high burst frequency or large bursts did exhibit fewer binary splicing observations (*Figure 2—figure supplement 1c*), but a linear regression showed that burst frequency and size did not contribute to binary observations beyond the effect of read coverage (*Figure 2—figure supplement 1d*). This suggests that transcriptional bursting contributes to splicing variability only by determining the overall abundance of an mRNA, consistent with single-molecule fluorescence observations (*Waks et al., 2011*).

### Simulations of RNA sequencing reveal technical sources of distortion of splicing estimates

A simple probabilistic exercise shows the potential loss of splicing information during sequencing. Single cell RNA-seq experiments that capture full-length transcripts have an estimated capture efficiency of only ~10%, due to RNA degradation and inefficient reverse transcription (*Marinov et al., 2014*; *Grün et al., 2014*; *Ziegenhain et al., 2017*; *Qiu et al., 2017*). For instance, a gene that expresses 20 mRNA molecules in a cell might only have two mRNAs recovered, and if that gene is

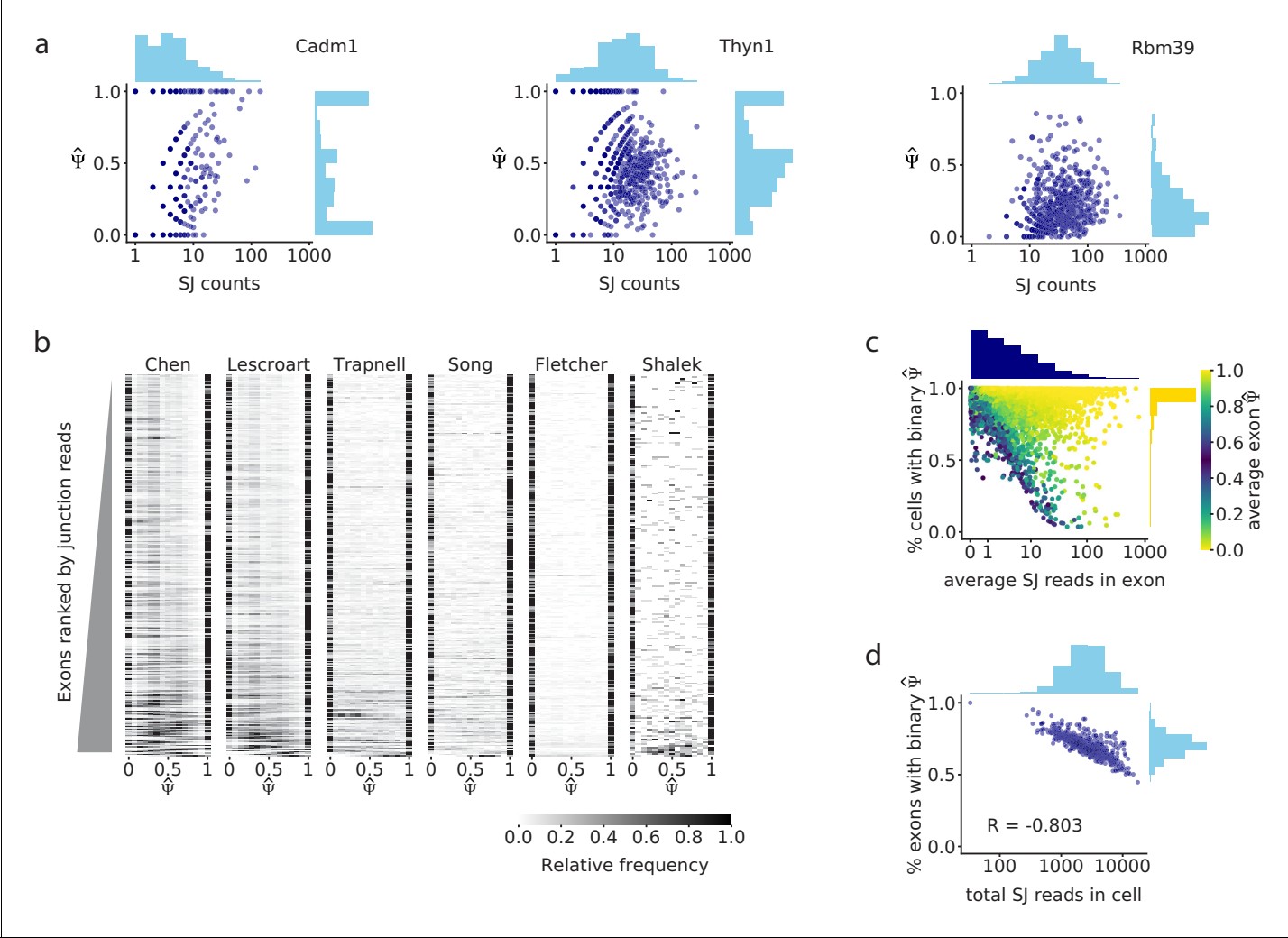

**Figure 2.** Splice junction read coverage is correlated with unimodality of splicing distributions. (a) Comparison of splice junction read coverage and observed $\hat{\Psi}$ for three cassette exons in the Chen dataset, with low (Cadm1 exon 8; chr9: 47829377–47829409), medium (Thyn1 exon 6, chr9: 27006801–27006951), and high coverage (Rbm39 exon 3; chr2: 156178880–156178952). Each dot represents the $\hat{\Psi}$ of that exon in one cell. (b) $\hat{\Psi}$ distribution of the 300 highest coverage cassette exons with intermediate splicing (average $\hat{\Psi}$ between 0.2 and 0.8) in each of the six analyzed datasets. Each row in the heatmap shows the distribution of $\hat{\Psi}$ for one exon across all cells. (c) Relationship between the average read coverage and proportion of binary observations for each cassette exon in the Chen dataset. (d) Correlation between the total number of splice junction reads captured in each cell, and proportion of cassette exons with intermediate splicing that show binary $\hat{\Psi}$ in that cell.

The online version of this article includes the following figure supplement(s) for figure 2:

**Figure supplement 1.** Junction read coverage determines the proportion of binary observations in all analyzed datasets.

alternatively spliced with a true splicing rate $\Psi$ of 0.5, there is approximately a 50% chance that those few recovered mRNAs will only represent one of the two isoforms that were originally present in the cell (*Figure 3—figure supplement 1a,b*). As many genes are expressed at just a few RNA molecules per cell, low recovery might affect many alternative splicing events (*Shapiro et al., 2013*; *Zenklusen et al., 2008*). Furthermore, while the empirically observed $\hat{\Psi}$ provides a maximum likelihood estimate for the true splicing rate, the uncertainty of this estimate (i.e. the range of alternative values with a nearly similar likelihood) decreases substantially with the number of observed molecules (*Figure 3—figure supplement 1c* and Materials and methods). As a result, the probability that the observed $\hat{\Psi}$ is close to the true underlying $\Psi$ increases when more mRNA molecules are captured (*Figure 3—figure supplement 1d*).

Our theoretical reasoning above relied on a simple model where the number of observed mRNA molecules (rather than number of reads) is known and the only distorting factor is a limited capture efficiency. In practice, both of these assumptions are challenged due to additional factors, such as PCR amplification and variability in the capture efficiency across cells. To investigate the pertinent effects on $\hat{\Psi}$ distributions in this more complex setting, we designed a probabilistic simulator of alternative splicing in single cells (*Figure 3—figure supplement 2*). The model has two main components: we begin by simulating the underlying molecular content of each cell, by drawing gene expression levels and cassette exon splicing rates from a probabilistic model of cell state. We then simulate the technical process of extracting data from each cell using single-cell RNA sequencing with full transcript coverage. This part accounts for variability in capture rates, and the effects of PCR amplification, fragmentation and sequencing. It relies on SymSim, a simulation tool for single-cell RNA sequencing data (*Zhang et al., 2019*). The final product of our simulation is the number of splice junction reads that either span or skip each exon in each cell. These numbers are distorted in a way that reflects real nuisance factors. For instance, two reads could have originated from the same molecule due to amplification effects.

We used our simulator to investigate how the observed inclusion ($\hat{\Psi}$) of cassette exons differs from the underlying $\Psi$, under different average capture rates, and setting the other technical parameters to values that are characteristic of Smart-seq2 datasets (see Materials and methods). We considered either a binary-bimodal regime of $\Psi$ (i.e. both isoforms are expressed in the population, but rarely by the same cell; *Figure 1a*), or a non-binary regime (cells tend to express both isoforms; *Figure 1b*). We simulated splicing of cassette exons in 1500 genes, in a population of 300 single cells. In the bimodal simulation, 500 exons were modeled to have alternative splicing with a bimodal distribution across cells, and in the unimodal simulation, these 500 exons were modeled to have a unimodal distribution. To reflect the patterns seen in real data, we also simulated splicing of 500 exons that were nearly constitutively included and 500 that were nearly constitutively skipped.

As expected, in the binary-bimodal simulation, the observed $\hat{\Psi}$ reflected the underlying process well, independent of the average capture efficiency (*Figure 3a*). In contrast, when we modeled a non-binary splicing regime, the observed $\hat{\Psi}$ distributions were strikingly similar to the splicing distributions of cassette exons in real single-cell RNA-seq datasets (*Figure 3b*). Specifically, the loss of information due to mRNA recovery and library generation led many of the observed $\hat{\Psi}$ to become binary, and their observed distribution across cells to become bimodal. This tendency again correlated with coverage, whereby lowly covered exons showed the strongest effect, while exons with high coverage maintained a non-binary, unimodal distribution. Consistently, in this non-binary regime, the average of $\hat{\Psi}$ was similar to the true average of $\Psi$, but the variance of $\hat{\Psi}$ increased (*Figure 3—figure supplement 1e,f*). Furthermore, as in the real data sets (*Figure 2c*), we also found that the dependency between read coverage and the chance of observing a binary $\hat{\Psi}$ is more pronounced in exons with an underlying $\Psi$ that is far from binary (*Figure 3c–f*), highlighting again that such an association likely indicates an artifact.

To address the extent of the distortion from mRNA recovery, we ran the simulator with varying capture efficiency and a fixed underlying $\Psi = 0.5$. We found that decreasing the average capture efficiency dramatically increased the number of binary $\hat{\Psi}$ observations, particularly for exons with low expression, although even highly expressed genes suffered great distortion in the observed $\hat{\Psi}$ when the average capture efficiency was very low (*Figure 3g*). These results reinforced the hypothesis that capture efficiency is a main technical factor that creates the appearance of bimodality in single-cell splicing.

Finally, we estimated the chance of observing only one type of isoform in any given cell (i.e. a binary $\hat{\Psi}$) as a function of the underlying $\Psi$ and the number of transcripts that are present in the cell (*Figure 3h*). For this analysis, we set an average mRNA capture rate of 10%. Our results delineate the range of values in which an artifact is less expected. For instance, for an exon with 50% inclusion rate, a non-binary estimate is more likely if the respective gene has at least 50 transcripts in the cell. Notably, these estimates are more conservative than the theoretical analysis (*Figure 3—figure supplement 1a,b*), due to the effect of the technical nuisance factors we modeled in this simulation.

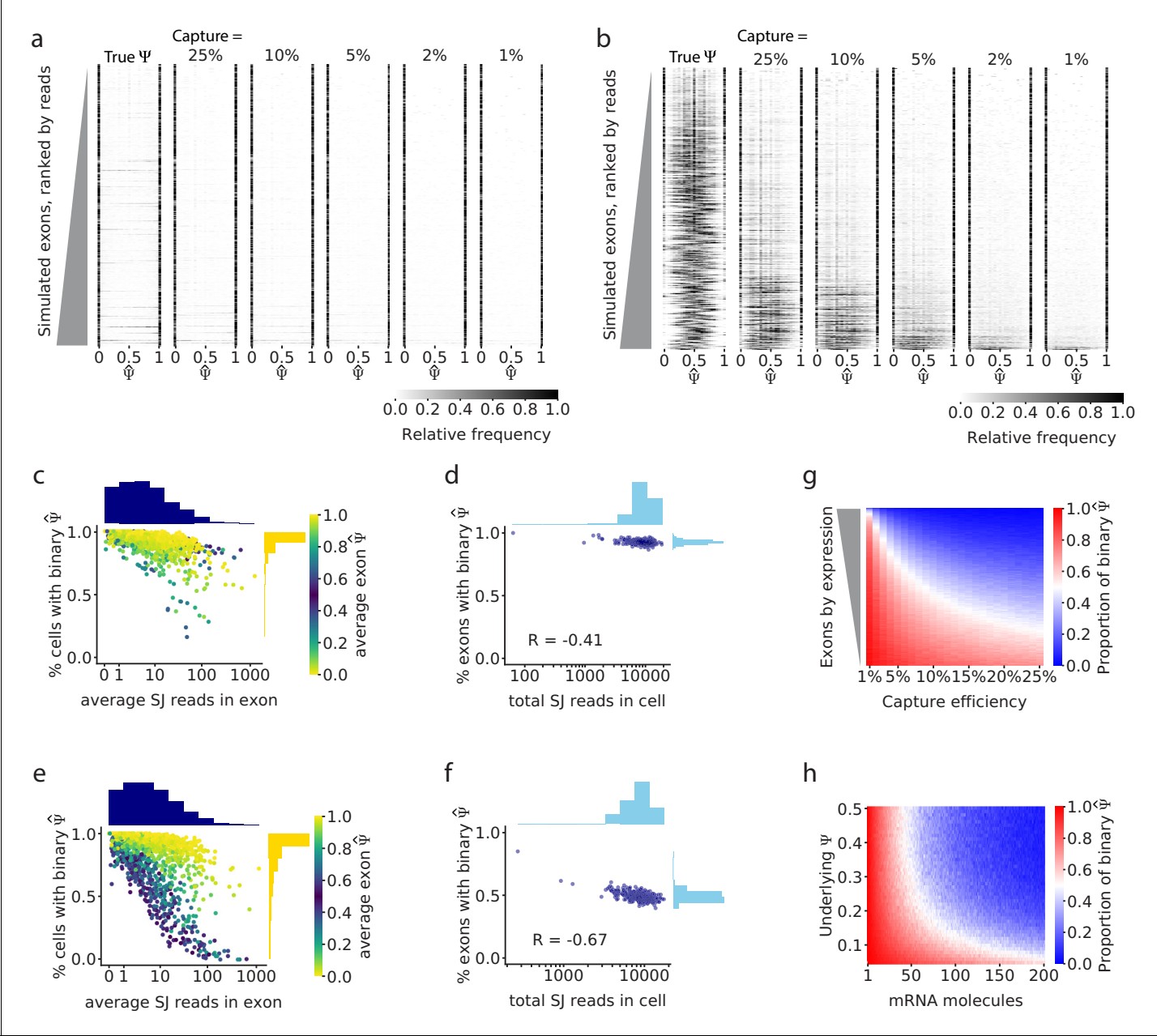

**Figure 3.** Simulations show that gene expression and capture efficiency influence the observed distribution of splicing. (a) Simulations of alternative splicing and scRNA-seq under the binary-bimodal model, in which each cell produces one isoform or the other, but rarely both. As in **Figure 2b**, each row of the histogram shows $\hat{\Psi}$ for one intermediate exon across all cells. The observed $\hat{\Psi}$ distribution is similar to the true $\Psi$ distribution, and its shape is largely unaffected by capture efficiency. (b) Simulations with the non-binary, unimodal model, in which most cells present a mixture of the two alternative isoforms. Exons with high expression have a unimodal distribution of true $\Psi$. Low capture efficiency results in an increase in binary observations (only one isoform observed), leading to a distortion of the observed distribution of $\hat{\Psi}$ to look bimodal. Only a handful of the highest expressed exons maintain a unimodal distribution of $\hat{\Psi}$. Fewer exons show unimodal splicing as the capture efficiency is reduced. (c) Under the binary-bimodal model, exons with high coverage have slightly fewer binary $\Psi$ observations, and (d) simulated cells with a high number of total splice junction reads have slightly fewer exons with binary $\hat{\Psi}$. (e) Under the unimodal model, exons with intermediate splicing show a strong decrease in binary observations as coverage increases, as seen in real data (**Figure 2c**). (f) Similarly, simulated cells with high read coverage have a decrease of the proportion of binary $\hat{\Psi}$. (g) Effect of capture efficiency on the proportion of binary observations of cassette exons with underlying $\Psi = 0.5$. (h) Effect of the initial number of mRNA molecules and underlying $\Psi$ on the proportion of binary $\hat{\Psi}$ observations.

The online version of this article includes the following figure supplement(s) for figure 3:

**Figure supplement 1.** Theoretical calculations and simulations of the effect of biological and technical factors in splicing observations.

*Figure 3 continued on next page*

*Figure 3 continued*

**Figure supplement 2.** Schematic of scRNA-seq splicing simulator.

## Accounting for mRNA recovery improves analysis of alternative splicing

We sought criteria that would identify reliable measurements of splicing in single cells and avoid distortions from low mRNA recovery. While ideally we should rely on the actual number of mRNA molecules recovered, full-length RNA-seq experiments generally do not report an absolute mRNA count. Previous studies assessed the quality of splicing observations based on the number of reads covering alternative splice junctions (*Song et al., 2017*; *Linker et al., 2019*), but the number of splice junction reads is influenced by the extent of PCR amplification and sequencing depth, and so it does not directly reflect the number of recovered mRNAs.

We realized that we could estimate the number of mRNA molecules that were captured into cDNA by using the Census normalization approach proposed by *Qiu et al., 2017*. This method infers a per-cell scaling factor between the relative abundance of each transcript, inferred from RNA-seq, and the actual number of mRNAs recovered. We found that some datasets with many reads per cell, such as the Song dataset (*Song et al., 2017*), nonetheless had few mRNAs recovered per cell (*Figure 4a*, *Figure 4—figure supplement 1c*), which may explain the extensive splicing bimodality in this dataset. The dataset with the highest recovery of mRNAs (*Chen et al., 2016*) indeed showed less binary splicing (*Figure 2b*).

Next, we sought a threshold for mRNA recovery that would suppress spurious observations of binary splicing. Our simulations and our analytical calculations suggested that, for a capture efficiency of 10%, recovering 7–10 mRNA molecules would be more likely than not to capture both isoforms for exons with intermediate splicing (*Figure 3h*), and that the observed $\hat{\Psi}$ would most likely be within 0.1 of the real value (*Figure 2—figure supplement 1d*). In keeping with this, we saw in the real data that exons with an average of at least 10 mRNAs recovered per cell generally had substantially fewer binary observations (*Figure 4b*, *Figure 4—figure supplement 1d*; limited to exons with average $\hat{\Psi}$ between 0.05 and 0.95). Nonetheless, a subset of these exons still showed binary splicing in most cells. These exons had few splice junction reads relative to the estimated mRNA count. We expect that these represent anomalously low recovery of reads from the specific splice junctions of interest, perhaps due to annotation errors or poor recovery of fragments with particular sequence composition. We reasoned that a data-driven splice junction read criterion would prevent distortions arising from this low read recovery. To find an appropriate read minimum, we determined the number of splice junction reads expected to arise from a cassette exon in each cell in different datasets, based on that cell's overall mRNA recovery (*Figure 4c*; see Materials and methods). This provided a second filtering criterion: we excluded exons with fewer splice junction reads than would be expected from 10 mRNAs, given that exon's $\hat{\Psi}$ and the coverage rate in that particular cell. This metric is calculated for each exon separately, driven by the actual information in each cell, and it varies substantially between datasets.

We then selected the exons from each cell that met the combined requirement of 10 mRNAs and the splice junction reads that would arise from 10 mRNAs, and asked what effect this filter had on binary observations of splicing. As an example, our filter removed seemingly spurious binary observations for alternatively spliced exon 8 of Cadm1 (*Figure 4d*). The observations that remained after filtering showed a clear pattern of increased inclusion along pseudotime, with exon skipping in stem cells and exon inclusion in fully differentiated neurons (Spearman's $r_s$ = 0.52, p=1.2 $\times$ 10$^7$; *Figure 4e*). The correlation between splicing observations and differentiation pseudotime highlighted that cell type differences could contribute to observations of bimodality that are not the result of an artificial inflation of binary observations, and this was an important factor to consider in our analysis.

## Bimodality in filtered exons is explained by differential splicing between cell types

Earlier papers had reported a surprising amount of bimodal splicing within a cell type, with extensive binary outcomes in individual cells, that is similar cells with different dominant isoforms. This

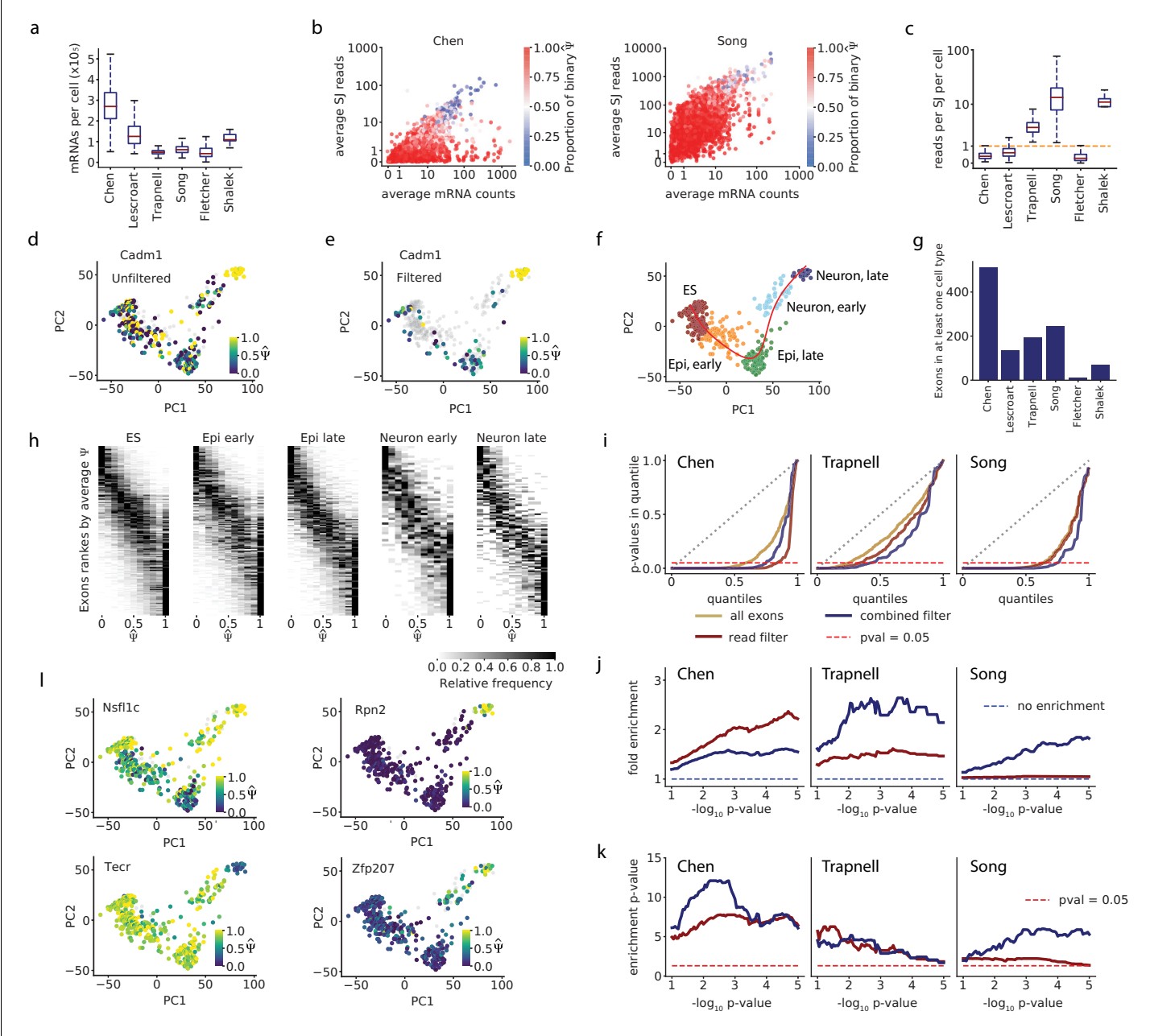

**Figure 4.** Accounting for coverage biases reveals unimodal splicing distributions and differential splicing. (**a**) Estimated total mRNA molecules captured per cell. (**b**) Estimated number of recovered mRNAs vs splice junction reads for cassette exons, averaged across cells. Each dot corresponds to an exon, and its color indicates the proportion of cells in which it has a binary observation (only one isoform observed). We analyzed exons with average $\hat{\Psi}$ between 0.05 and 0.95. (**c**) Per-cell splice junction coverage rate in each dataset. (**d**) Cadm1 exon 8 alternative splicing appears binary in many cells in the Chen dataset. Correlation with lineage pseudotime: Spearman's $r_s$ = 0.1. (**e**) Cadm1 exon after removing cells with fewer than 10 recovered Cadm1 mRNA molecules and fewer splice junction reads than expected from 10 mRNAs (grey). Spearman's $r_s$ = 0.52. (**f**) PCA projection and clustering of single cells in the Chen dataset, showing differentiation of mouse ES cells into neurons. Red line, lineage inferred with Slingshot. (**g**) Number of cassette exons with observations from at least 10 mRNA reads in at least 50% of cells in any cluster. (**h**) Stacked histograms showing the distribution of observed $\hat{\Psi}$ of exons as in (**g**), in each cell cluster of the Chen dataset. Observations with fewer than 10 mRNA molecules were removed. We show exons with average $\hat{\Psi}$ ranging from 0.1 to 0.9 per cluster. (**i**) QQ-plot comparing the quantiles of a uniform distribution (x-axis) with the quantiles of the distributions of p-values from the Kruskal-Wallis test (y-axis). A diagonal line (gray dotted line) would mean the p-values are uniformly distributed. A lower area under the curve indicates enrichment for low p-values. The point on the x-axis at which each line crosses the dotted red line indicates the proportion of p-values that are below 0.05 in the distribution. (**j**) Fold enrichment of exons with a Kruskal-Wallis $p < x$ in the set of exons selected with the mRNA-based filter (blue), and exons selected with a flat read minimum filter (red). (**k**) Significance p-value of the enrichment, estimated with the

*Figure 4 continued on next page*

*Figure 4 continued*

hypergeometric test and adjusted for FDR. (I) Example exons that pass the overall filter criteria in the Chen dataset and have p<0.05 in the Kruskal-Wallis test.

The online version of this article includes the following figure supplement(s) for figure 4:

**Figure supplement 1.** Relationship between read coverage, captured mRNAs, and binarity in single cell datasets.
**Figure supplement 2.** Analysis of differential splicing among selected exons with the Kruskal-Wallis analysis of variance.
**Figure supplement 3.** Analysis of differential splicing among selected exons with the autocorrelation test.
**Figure supplement 4.** Distribution of $\hat{\Psi}$ of example exons.

suggested that some aspects of gene expression stochastically locked individual cells into producing primarily molecules of one or the other isoform. However, our results so far suggest that much of this observed variability may be an artifact of low mRNA recovery. In contrast, differences in splicing between cells of different types are consistent with conventional mechanisms such as regulation by cell-type-specific splicing factors. Therefore, when considering heterogeneous cell samples, large shifts in splicing between cell types may result in truly bimodal $\Psi$. In contrast to the first scenario, we would expect the corresponding observations $\hat{\Psi}$ to remain bimodal even after low-quality data points were removed.

We therefore sought to distinguish between these two scenarios and estimate the extent of bimodal splicing within a cell type that is strongly supported by the data. Most of the datasets in our analysis came from differentiation time courses, and so to examine homogeneous cell subsets, we stratified the cells into groups indicative of their developmental stage. The groups were identified by clustering the cells using the normalized sum of reads from every gene (i.e. not directly representing their variation in splicing; *Figure 4f*) and labeled based on expression of known marker genes (*Figure 4—figure supplement 1e*).

We found that, prior to filtering, hundreds of exons with intermediate splicing (average $\hat{\Psi}$ between 0.2 and 0.8) seemed to have at least weakly bimodal splicing distributions (using a heuristic definition of at least 25% of cells with $\hat{\Psi} \leq 0.25\%$ and 25% with $\hat{\Psi} \geq 0.75$), a rate that roughly matched previous descriptions (*Song et al., 2017*; *Table 1*). We then discarded the observations that did not meet our filtering criteria, and retained for every cluster only those exons with observations remaining in at least 50% of cells (*Figure 4g*). With that filtering we found that most remaining exons had fewer binary observations than the discarded exons (*Figure 4—figure supplement 1f*).

**Table 1.** Prevalence of bimodal splicing distributions before and after filtering.

| Dataset | Cell type cluster | Cells | Exons | Bimodal | % bimodal | Selected | Selected bimodal | % bimodal selected | p-val (adj) |
|---|---|---|---|---|---|---|---|---|---|
| Chen | ES | 217 | 446 | 118 | 26% | 94 | 0 | 0% | 3.3e-14 |
| Chen | Epi, early | 98 | 402 | 107 | 27% | 98 | 1 | 1% | 9.3e-14 |
| Chen | Epi, late | 104 | 516 | 136 | 26% | 76 | 1 | 1% | 1.1e-09 |
| Chen | Neuron, early | 47 | 364 | 117 | 32% | 43 | 0 | 0% | 3.6e-08 |
| Chen | Neuron, late | 22 | 517 | 146 | 28% | 61 | 0 | 0% | 1.1e-09 |
| Lescroart | Heart E6.75 | 172 | 286 | 77 | 27% | 33 | 0 | 0% | 2.0e-05 |
| Lescroart | Heart E7.25 | 341 | 291 | 78 | 27% | 36 | 0 | 0% | 8.0e-06 |
| Trapnell | Myoblast 00 hr | 35 | 400 | 142 | 36% | 41 | 0 | 0% | 1.2e-08 |
| Trapnell | Myoblast 24 hr | 89 | 251 | 97 | 39% | 27 | 0 | 0% | 1.2e-06 |
| Trapnell | Myoblast 48 hr | 72 | 242 | 97 | 40% | 32 | 1 | 3% | 9.1e-07 |
| Trapnell | Myoblast 72 hr | 35 | 252 | 101 | 40% | 24 | 1 | 4% | 5.1e-05 |
| Song | iPSC | 62 | 616 | 269 | 44% | 55 | 0 | 0% | 3.3e-14 |
| Song | NPC | 73 | 212 | 92 | 43% | 28 | 1 | 4% | 1.2e-06 |
| Song | MN | 67 | 168 | 82 | 49% | 19 | 4 | 21% | 9.4e-03 |
| Shalek | BMDC | 13 | 149 | 51 | 34% | 27 | 1 | 4% | 6.6e-05 |

Furthermore, cell clusters had no or very few remaining exons with bimodal splicing distributions (*Figure 4h*, *Figure 4—figure supplement 1g,h*, *Table 1*). For instance, in the Chen dataset, there were two remaining cases of bimodality, and both seem to be the result of sex-specific splicing (*Figure 4—figure supplement 1i*). This trend persisted when reanalyzing all datasets and clusters using a range of cutoffs for qualifying a splicing pattern as binary (*Figure 4—figure supplement 1e*).

## Filtering improves the identification of differentially spliced exons

Finally, we asked if our filtering criteria could help identify biologically relevant changes in splicing between cell types, by selecting exons whose biological variance is not overwhelmed by high technical variance. Focusing on three datasets with multiple cell type clusters (Chen, Trapnell, and Song), we examined the effect of retaining only exons that passed our filter in at least 50% of cells in each cluster. To avoid biases depending on the number of observations available for each remaining exon, we retained all data points for these exons, rather than removing individual cell data points that did not pass the filter. As a benchmark for comparison, we used a simpler criterion, akin to previous studies, of at least 10 splice junction reads in at least 50% of cells (*Figure 4i*). To focus on exons with substantial alternative splicing, we omitted exons from this analysis that were consistently excluded (average $\hat{\Psi} < 0.05$) or included (average $\hat{\Psi} > 0.95$) across all cells.

Taken together, these results challenge the idea that widespread bimodality among homogeneous cells arises from a binary nature of splicing in single cells, and instead suggest that splicing bimodality reflects biological differences between cell types or subtypes. Indeed, experimental observations of splicing bimodality support the importance of differences between cell types, rather than stochastic differences between homogeneous cells. The two cases of splicing bimodality confirmed with smFISH by *Song et al., 2017* both showed bimodality between two characterized cell types. Similarly, *Shalek et al., 2013* confirmed one case of splicing bimodality, in a gene whose expression they showed to be a marker of a hidden difference between mature and maturing cells, and so its splicing may reflect cell state difference at a wider context as well.

For each remaining exon in the two filtering schemes, we performed a Kruskal–Wallis one-way analysis of variance, which assigned it a p-value (Materials and methods). This test determines if the exon's median $\Psi$ changed significantly between any clusters. Both filtering schemes enriched substantially for exons that show significant changes across different clusters, compared with the unfiltered set of all exons with average $\hat{\Psi}$ between 0.05 and 0.95. Furthermore, the extent of enrichment increased with the strictness of the differential splicing test (*Figure 4i–k*; *Figure 4—figure supplement 2*). It is interesting to note that in the Chen dataset, the baseline (ten reads) filter was stricter than the combined filter, retaining 66 vs. 198 exons and thus achieving high precision at the expense of a lower recall. Indeed, due to the high mRNA recovery and low amplification in this dataset, our combined filter required only seven splice junction reads, rather than ten as in the baseline scheme. Conversely, in the Song dataset, which is predicted to have low mRNA recovery yet a large number of reads, the baseline filter retained almost all exons, while the combined filter retained only 104 of them, leading to significant over-representation of exons that are differentially spliced between clusters. The results show how differences in mRNA recovery and amplification between datasets may change the interpretation of an absolute number of reads. We corroborated these results using spatial autocorrelation as a way of identifying informative exons, rather than the clustering and differential splicing analysis. The autocorrelation test builds on the work of *DeTomaso et al., 2019* to identify exons with a significantly high level of consistency in $\hat{\Psi}$ among transcriptionally similar cells (*Figure 4—figure supplement 3*).

Taken together, these results indicate that careful filtering of exons can help identify observations that are consistent with the gene expression space and the stratification of cells into types, and are thus likely more biologically meaningful (e.g. capturing known forms of regulation during differentiation) (*Figure 4l*, *Figure 4—figure supplement 4*; *Liu et al., 2018*). The concept of 'meaningful' is ascribed here to the observations, not the exons themselves, as there may be many additional exons with relevant variation (e.g. stable differences between cell types) that is not reliably measured due to low coverage. Indeed, the effects of the two filtering schemes and the differences between them exemplify how technical factors may distort our view of the data and consequently, our understanding of variation and regulation of splicing.

## Discussion

The surprising result that alternatively splicing is bimodal among single cells provoked curiosity and speculation. Bimodal outcomes might reflect hidden cell subtypes, but the bimodality was seen even among apparently homogeneous cells. Did splicing outcomes reflect some unknown, stochastic cell state?

We have shown here that the bimodal patterns could have an entirely different explanation: profound technical limitations of single cell RNA sequencing. A crucial limit on biologically meaningful splicing observations in a single cell is the number of mRNAs available to inform the measurement. This is determined both by the expression of the genes that contain the exons of interest, and by the capture efficiency of the experiment. It is important to note that the depth of a sequencing library does not necessarily reflect its quality. Along with low mRNA numbers, splicing observations are also distorted by uneven amplification efficiency and cDNA overamplification. Increasing PCR amplification cycles in an attempt to compensate for low capture efficiency has the risk of worsening the technical distortion. Indeed, in our analysis, the dataset with the highest read count per cell actually had quite low mRNA recovery and large technical distortion, creating an appearance of bimodal splicing (*Song et al., 2017*). Moreover, a qualitative change in the observed $\hat{\Psi}$ distribution of an exon between single cell subpopulations does not necessarily reflect a change in the underlying splicing rate, as changes in gene expression and mRNA recovery between samples can create the illusion of a splicing change.

Further developments in statistical analysis that carefully account for both missing and redundant information due to low capture efficiency could make splicing observations in single cells more reliable. We set the foundation for such analysis by proposing a probabilistic process that describes the biological and technical steps that generate single cell splicing data. We also introduced a simple approach that builds on the Census normalization (*Qiu et al., 2017*) to estimate the number of mRNAs recovered and the extent of artificial duplication of splicing information. This metric provides a practical filter for identifying exons with sufficient information to analyze. On the experimental side, improving the capture efficiency of scRNA-seq methods while moderating the extent of overamplification is crucial for increasing the subset of exons for which reliable observation can be made.

True biological insight into alternative splicing can indeed be found from high-quality scRNA-seq data, and we hope that new methods will allow better understanding of splicing regulation, cell-to-cell variation, and the importance of alternative splicing in defining cell fate (*Hagemann-Jensen et al., 2020*; *Gupta et al., 2018*). However, some limitations are inherent to the situation. Single cells express a limited number of mRNAs per gene; splicing observations in single cells will always be inherently noisy reflections of the underlying biology.

## Materials and methods

Analysis code is available at https://github.com/lareaulab/sc_binary_splicing. (*Buen Abad Najar and Lareau, 2020*; copy archived at https://github.com/elifesciences-publications/sc_binary_splicing).

### Analysis of single-cell RNA-seq datasets
#### Datasets
Six publicly available single cell RNA sequencing datasets were analyzed (*Table 2*). These datasets are referenced with the first author's name throughout this paper. For the Chen dataset, we selected the four cell types used to represent developmental stages in *Chen et al., 2016*: mouse embryonic stem cells cultured in 2i (ES2i) and LIF (ES), mouse EpiStem cells (Epi) and neurons. For the Lescroart dataset, we limited the analysis to the cells derived from mouse wild-type strains. For the Trapnell dataset, we only selected the runs that are annotated to have one cell per well.

### Alignment, TPM quantification and $\Psi$ estimation
We aligned the reads of each dataset using STAR 2.5.3 (*Dobin et al., 2013*) with two-pass mode and index overhang length adapted to the read length of each dataset. We used the hg38 genome annotation for the human RNA-seq datasets, and the mm10 annotation for the mouse datasets. Gene expression levels in transcripts per million (TPM) were calculated by running RSEM (*Li and Dewey, 2011*) on the BAM files produced by the STAR alignment. We ran rMATS 3.2.5 (*Shen et al.,*

**Table 2.** Single-cell RNA-seq datasets.

| Dataset | Organism | Biological process | Cells | No. ASE | Mean reads/event | Protocol | Accession | Reference |
|---|---|---|---|---|---|---|---|---|
| Chen | mouse | mES neuron differentiation | 488 | 3276 | 3.1 | Smart-seq2 | GSE74155 | *Chen et al., 2016* |
| Lescroart | mouse | cardiomyogenesis | 598 | 3007 | 3.2 | Smart-seq2 | GSE100471 | *Lescroart et al., 2018* |
| Trapnell | human | skeletal myogenesis | 314 | 4457 | 14.4 | Smart-seq | GSE52529 | *Trapnell et al., 2014* |
| Song | human | iPS motor neuron differentiation | 206 | 5355 | 88.8 | Smart-seq | GSE85908 | *Song et al., 2017* |
| Fletcher | mouse | olfactory neurogenesis | 849 | 684 | 2.7 | Smart-seq | GSE95601 | *Fletcher et al., 2017* |
| Shalek | mouse | bone-marrow-derived dendritic cells | 18 | 380 | 213.4 | Smart-seq | GSE41265 | *Shalek et al., 2013* |

*2014*) on bulk human and mouse RNA-seq datasets from cell types matching the scRNA-seq datasets (*Chen et al., 2016*; *Hubbard et al., 2013*; *Busskamp et al., 2014*; *Trapnell et al., 2014*) to find all annotated cassette exon alternative splicing events in each cell type. Then we used the SJ.out.tab files obtained from the scRNA-seq STAR alignment to obtain the splice junction reads compatible with the list of cassette exons found by rMATS. For each cell $i$, we calculated the observed $\hat{\Psi}$ of the cassette exon $j$ as:

$$\hat{\Psi}_{ij} = \frac{SJ_{A_{ij}}}{SJ_{A_{ij}} + 2SJ_{B_{ij}}}$$

where $SJ_{A_{ij}}$ correspond to the number of reads that cover the two splice junctions compatible with cassette exon inclusion, and $SJ_{B_{ij}}$ are the reads that cover the splice junction compatible with its exclusion. We also determined the coverage of an exon $j$ in $i$ as $SJ_{ij} = SJ_{A_{ij}} + SJ_{B_{ij}}$. We used $SJ_{ij}$ and $\hat{\Psi}_{ij}$ for the analyses shown in *Figure 2*.

## Gene expression normalization and pseudotime inference

For the purpose of visualization and clustering of cells, we normalized the gene expression data. First, we selected the genes with TPM > 20 in at least 20 cells. After filtering, we used SCONE 1.6.1 (*Cole et al., 2019*) to select the best normalization approach for the data. For improving the normalization of the data, we used additional information for each cell, including the annotated cell type and batch, total number of reads, housekeeping genes and genes that are expected to change in the biological process that the dataset covers. We applied principal component analysis (PCA) over the log-counts from the best SCONE normalization, and used the first two principal components to infer pseudotime using Slingshot 1.0.0 (*Street et al., 2018*). We used the cell type annotation as the cluster input for Slingshot, and manually indicated the direction of the biological process. Single cells can be highly heterogeneous, even within cells labeled to be in the same biological condition. To account for this variation, instead of relying on the annotated cell type provided with each dataset, we used agglomerative clustering over the PCA projection of the matrix of normalized gene expression to identify groups of similar cells in the Chen, Trapnell and Song datasets. For the Trapnell and the Song dataset, we used the number of cell types reported by the authors. In the Chen dataset, we noticed that the cells annotated as neurons could form two well-defined clusters, with some of the cells presenting expression profiles consistent with mature neurons. For this reason, we used five instead of four clusters in the Chen dataset. In the Lescroart dataset, we used the original labels provided by the authors because PCA over the normalized expression data separates the cells into two groups. This analysis was not performed in the Shalek dataset due to its small number of cells. Instead, we treated all the cells in this dataset as one single cluster. The Fletcher dataset was not considered for the analysis that required clustering due to the low number of exon observations that pass the minimum mRNA requirements described below.

## mRNA counts estimation with the Census approach

We performed our own implementation of the mRNA count estimation proposed in Census (*Qiu et al., 2017*). The total number of transcript mRNAs in cell $i$ is estimated as:

$$M_i = \frac{n_i}{F_{X_i}(x_i^*) - F_{X_i}(0.1)}$$

where $x_i^*$ is the mode of the log-transformed distribution of TPM values in cell $i$. As in Census, we found $x_i^*$ by fitting a Gaussian kernel density estimation to each distribution and finding its peak. We also set 0.1 as is the minimum TPM below which it is assumed that no mRNA is present. $n_i$ is the number of genes in cell $i$ with an estimated TPM in the interval $(0.1, x_i^*)$. $F_{X_i}$ is the cumulative distribution function of the TPM values in cell $i$. The original Census implementation also adjusts the mRNA estimation by multiplying by $\frac{1}{\theta}$, where $\theta$ is the capture efficiency of the dataset estimated with RNA spike-ins. Since for most datasets, we do not have a reliable way of estimating the capture efficiency, we removed this adjustment from the equation, so that $M_i$ in our estimation is not an estimation of the amount of mRNAs present in the cell lysate as it is in Census, but an estimate of the mRNAs successfully captured into cDNA.

We found that some datasets contained outlier cells with $M_i$ much higher than the median estimate (more than ten-fold increases). These outliers generally correspond to cells with a multimodal TPM distribution. An inflation in very low TPM values distorts the normalization by shrinking the values of $F_{X_i}(x_i^*) - F_{X_i}(\epsilon)$, thus inflating the $M_i$ values in these cells. Because the Census method relies on a Gaussian kernel density estimation that performs inaccurately for multimodal distributions, we excluded this handful of outliers from further analysis. In the Trapnell dataset, we found that several cells had an unusually small number of recovered reads and genes observed. We reasoned that this had the potential to skew the TPM estimations on which the Census normalization depends (*Figure 2—figure supplement 1b*), so we excluded the cells in the bottom 25% quantile of reads from the Census normalization and all downstream analyses that depend on it.

Finally, the number of mRNA transcripts of gene $g$ in cell $i$ is calculated as:

$$Y_{ig} = X_{ig} \cdot \frac{M_i}{10^6}$$

where $X_{ig}$ is the expression of gene $g$ in cell $i$ expressed in TPM.

## Nucleotide coverage and expected splice junction reads

Amplification in short-read library preparation can lead to multiple reads from the same sequence fragment of a single mRNA molecule. To filter out exons with anomalously low read coverage, we wanted to know the number of splice junction reads expected to originate from one exon junction in a single mRNA, a number which differs in each cell and each experiment. We estimated the splice junction coverage rate of each cell as the expected number of reads covering the splice junction of a mRNA molecule:

$$
\begin{aligned}
C_j \ &= \text{read coverage at each splice junctio of mRNAs in our sample} \\
&= \frac{\text{constitutive splice junction reads in cell } j}{\text{constitutive splice junctions of mRNAs in cell } j} \\
&= \frac{\sum_k r_{jk}}{\sum_k j_k \cdot m_{jk}}
\end{aligned}
$$

where $C_j$ is the splice junction coverage rate, which is the number of reads expected to cover each splice junction in cell $j$; $r_{jk}$ is the number of reads that map to all constitutive splice junctions of gene $k$ in cell $j$, as reported by STAR; $j_k$ is the number of constitutive splice junctions in a mRNA molecule of gene $k$; and $m_{jk}$ is the Census estimate of captured mRNAs of gene $k$ in cell $j$. To calculate the $C_j$ of each cell, we only considered the constitutive splice junctions of genes that were estimated to be expressed in at least one mRNA molecule by the Census normalization. This calculation captures the slight under-counting of splice junctions (relative to other positions) because of factors including ambiguous read mapping.

We identified the constitutive splice junctions of the human and mouse transcriptome using the Gencode hg38 and mm10 annotations respectively. For each organism, we identified the splice junctions that occur in all protein coding isoforms of each protein coding gene. This

resulted in a total of 59,477 constitutive splice junctions in the human genome, and 108,481 in the mouse genome.

Then, based on the overall recovery of splice junction reads in a cell, the total expected splice junction reads for a particular cassette exon $i$ in cell $j$ is estimated as:

$$\mathrm{SJ}_{E_{ij}} = m_{ij} \cdot (\text{expected splice junctions per mRNA}) \cdot C_j$$
$$= m_{ij} \cdot (1 + \hat{\Psi}_{ij}) \cdot C_j$$

where $SJ_{E_{ij}}$ is the expected number of splice junction reads covering the splicing of exon $i$ in cell $j$ (both for mRNAs that splice in or skip the exon). $m_{ij}$ is the estimated number of mRNAs from the gene containing the cassette exon $i$ in cell $j$; $\Psi_{ij}$ is the observed splicing rate of exon $i$ in cell $j$. The expected number of splice junctions per mRNA is $1 + \hat{\Psi}_{ij}$ because one splice junction read is present in mRNA molecules that skip the exon, and two in those that include it.

## Filtering to select good quality observations

Simulations of the effect of the initial number of mRNA molecules of a gene and the underlying $\Psi$ suggest that an average of 44 mRNA molecules are necessary to have a 50% chance of making an intermediate $\Psi$ observation when the underlying $\Psi$ is 0.5. This number goes up to 65 if the $\Psi$ is 0.2 or 0.8, and to 127 if the $\Psi$ is 0.1 or 0.9 (*Figure 4—figure supplement 1f*). Assuming a capture efficiency of 10%, we rounded at 10 captured mRNA molecules as the lower threshold for a quality $\Psi$ observation.

In some cases, the number of observed splice junction reads is discordant with the estimated number of mRNAs recovered. Therefore, we set a additional filter based on the number of reads expected to come from 10 mRNA molecules that are informative about the splicing of a cassette exon:

$$SJ_{m_{ij}} = 10 \cdot (1 + \hat{\Psi}_{ij}) \cdot C_j$$

Therefore, for every observation, we required at least 10 mRNAs of the gene captured, and at least the number of reads that we expect if 10 mRNAs are informative. Notice that this minimum will be unique to each observation (combination of cassette exon and cell), as it depends on the cell-specific coverage rate, and the cell and exon specific observed $\hat{\Psi}$.

## Kruskal-Wallis test and filter evaluation

We evaluate the significance of splicing changes between different cell types, using the clusters defined above, for the Chen, Trapnell, and Song datasets. For each exon, we asked if its median $\hat{\Psi}$ is different between clusters. In order to do this, we grouped the exon's $\hat{\Psi}$ observations by cluster and ran the Kruskal–Wallis test, which is a non-parametric one-way analysis of variance across all clusters at once. We reported the p-value from each exon. A significant result indicates that the exon has a significantly different median $\hat{\Psi}$ in at least one cluster relative to at least one other cluster, indicating cell type-associated differential splicing. A non-significant result from the test suggests that the exon's median $\hat{\Psi}$ is not significantly different between any pairs of clusters. The strictness of the differential splicing test was determined by setting different p-value thresholds.

## Autocorrelation test

We reasoned that exons with reliable estimation of their splicing rate would tend to have similar observations ($\hat{\Psi}$) in cells that are transcriptionally similar to each other. To quantify this, we adapted the autocorrelation test described in *DeTomaso et al., 2019* to compute, for every exon, the similarity in $\hat{\Psi}$ amongst neighboring cells in the space of the top two principal components of the gene expression space.

To calculate the autocorrelation score of one exon, first we normalized the $\hat{\Psi}$ as follows:

$$\hat{\Psi}'_{ij} = \frac{\hat{\Psi}_{ij} - \overline{\Psi}_j}{\mathrm{Var}(\hat{\Psi}_j)}$$

where $\hat{\Psi}'_{ij}$ is the normalized $\hat{\Psi}_{ij}$ of exon $i$ in cell $j$; $\overline{\Psi}_j$ is the average observed $\hat{\Psi}$ of all the tested exons in cell $j$, and $Var(\hat{\Psi}_j)$ is the variance of all observed $\hat{\Psi}$ in cell $j$.

For each cell $j$, we identified all its $k$-nearest neighbors in the PCA projection of the normalized gene expression. For each neighbor $k$, we calculated a similarity score as follows:

$$w_{jk} = \exp\left(\frac{-d_{jk}^2}{\sigma_j^2}\right)$$

where $d_{jk}$ is the Euclidean distance in the PCA projection between cell $j$ and its neighbor $k$; $\sigma_j$ is the Euclidean distance from cell $j$ to its farthest $K$-nearest neighbor. For all cells $k$ that are not neighbors of $j$, we set $w_{jk} = 0$. For each analyzed dataset, we set $K$ to be half the total number of cells. The smart-seq and smart-seq2 datasets that we analyzed here contain orders of magnitude fewer cells than the UMI-based datasets analyzed by DeTomaso et al., and we reasoned that the default setting of $K = \sqrt{N}$ might result in neighborhood sizes too small to ensure stability in some datasets. In the datasets with fewer cells, $K = \frac{N}{2}$ agreed better than $K = \sqrt{N}$ with the results of the Kruskal-Wallis test (Jaccard indexes: 0.52 and 0.45 respectively in the Song dataset, 0.43 and 0.28 in the Trapnell dataset, and 0.51 and 0.54 in the Chen dataset).

For each exon $i$, we calculate the autocorrelation score as a variation of the Geary's C statistic as:

$$C' = 1 - \frac{(N-1)\sum_j \sum_k w_{jk}(\hat{\Psi}'_{ij} - \hat{\Psi}'_{ik})^2}{2W\sum_j (\hat{\Psi}'_{ij} - \overline{\Psi}'_i)^2}$$

where $N$ is the total number of cells in the dataset, and $W$ is the sum of all $w_{ij}$.

For each exon, we calculated an empirical p-value. To avoid bias from missing data and different splicing rates, we binned all the exons by average $\hat{\Psi}$ and percent of missing values. We grouped exons with average $\hat{\Psi}$ of: {(0.05−0.1 & 0.9−0.95), (0.1−0.2 & 0.8−0.9), (0.2−0.3 & 0.7−0.8), (0.3−0.4 & 0.6−0.7), (0.4−0.6)}. We also binned exons with missing observations between: {(50−60%), (60−70%), (70−80%), (80−90%), (90−100%)}.

For each bin, we randomized the values of the exons across the cells 20,000 times. We then calculated the autocorrelation score for all the randomized exons.

For each exon, we calculated its p-value as $p = \frac{x+1}{20,001}$, where $x$ are all the randomized exons of the same bin with a higher autocorrelation score.

## Splicing change analysis and filter evaluation

We evaluated if a subset of exons is enriched for exons with low p-values indicating significant change. We repeated this analysis for the p-values from the Kruskal-Wallis test and from the autocorrelation test. For each $x$ as a threshold of low p-values, we define:

1. $M$ as the set of all exons.
2. $m$ as the set of exons with p-value $\leq x$.
3. $P$ as the exons in the selected subset.
4. $p$ as the exons in the subset with p-value $\leq x$.

We then calculate the fold enrichment as follows:

$$\text{enrichment} = \frac{|M| \cdot |p|}{|P| \cdot |m|}$$

We then use the hypergeometric test to determine if the subset of selected exons is significantly enriched for exons with p-values below the threshold. In our analysis, we tested $x$ in a range between 0.1 and 0.00001, and corrected the p-values of the hypergeometric test for multiple testing using the Benjamini-Hochberg correction.

In addition, we define:

1. $p$ as true positives.
2. $\neg m$ AND $\neg P$ as true negatives.
3. $\neg m$ AND $P$ as false positives.

4. $m$ AND $\neg P$ as false negatives.

Using these definitions, we calculate the precision, recall, specificity, accuracy and F1 score of the filter for every $x$.

## Linear regression on transcriptional burst kinetics

Transcriptional burst kinetics parameters (burst frequency and burst size) of mouse embryonic stem cells were obtained from *Larsson et al., 2019*. The authors modeled these parameters from cells from the Chen dataset (*Chen et al., 2016*). We selected 619 intermediate exons observed in the ES2i cells that have binary observations (i.e., $\hat{\Psi}$ = 0 or 1) in between 1% and 99% of the cells. For each exon, we calculated the logit of the proportion of cells that present binary observations as the target variable. Additionally, we matched each exon to three predictive features: the transcriptional burst size of its gene, the transcriptional burst frequency of its gene, and its expression, represented as the average number of informative splice junction reads that cover the exon. Each predictive feature was transformed by $\log_{10}+1$, and all variables were scaled to its standard score. We trained a linear regression model to predict the logit of the proportion of binary exons, using several combinations of the previously described predictive features:

1. $\text{logit}(\Psi) \sim$ burst size
2. $\text{logit}(\Psi) \sim$ burst frequency
3. $\text{logit}(\Psi) \sim$ size + freq + size $\cdot$ freq
4. $\text{logit}(\Psi) \sim$ expression
5. $\text{logit}(\Psi) \sim$ size + freq + exp + size $\cdot$ freq + size $\cdot$ exp + freq $\cdot$ exp

We evaluated each model by calculating the $R^2$ score between the $\text{logit}(\Psi)$ and the regression prediction.

## Theoretical analysis of the observed $\hat{\Psi}$ with limited capture rate

mRNA molecules are captured at a limited rate, approximated in some instances as 10% of the molecules in the cell. Under the assumption of uniform sampling of transcripts and isoforms, and assuming the only nuisance factor is the limited capture rate, we formalize the probability for observing a splicing ratio $\hat{\Psi}$. We start by specifying this probability, assuming that we know the total number of transcripts from the respective gene in the cell ($m$), the real splicing rate $\Psi$ and the number of captured molecules $r$ (assuming that for any capture molecule we know if it includes the exon or not). In that case:

$$Pr(\hat{\Psi} \mid \Psi, r, m) = \frac{\binom{m\Psi}{r\hat{\Psi}}\binom{m(1-\Psi)}{r(1-\hat{\Psi})}}{\binom{m}{r}}$$

Note that for this calculation, the capture efficiency ($c$) is not needed, since we assume that we know $m$ and $r$. For a more useful analysis, we will next assume that only one of these variables is not known (starting with $m$ and then $r$).

In a more realistic scenario, $r$ and $c$ can be estimated (e.g. using Census), while $m$ remains unknown. We can therefore marginalize $m$ to calculate:

$$\begin{aligned}
\Pr(\hat{\Psi} \mid \Psi, r, c) &= \sum_{m=0}^{\infty} \Pr(\hat{\Psi}, m \mid \Psi, r, c) \\
&= \sum_{m=r}^{\infty} \Pr(\hat{\Psi} \mid \Psi, r, c, m) \cdot \Pr(m \mid r, c) \\
&= \sum_{m=r}^{\infty} \Pr(\hat{\Psi} \mid \Psi, r, m) \cdot \Pr(m \mid r, c)
\end{aligned}$$

To estimate $\Pr(m \mid r, c)$ we note the following:

$$\begin{aligned}
\Pr(m \mid r,c) &= \frac{\Pr(r \mid c,m) \cdot \Pr(m)}{\sum_{m'=0}^{\infty} \Pr(r \mid c,m') \cdot \Pr(m')} \\
&= \frac{\Pr(r \mid c,m)}{\sum_{m'=0}^{\infty} \Pr(r \mid c,m')} \\
&= \frac{\binom{m}{r} c^r (1-c)^{m-r}}{\sum_{m'=0}^{\infty} \Pr(r \mid c,m')}
\end{aligned}$$

where we model the probability of capturing $r$ mRNA molecules as a binomial sample from $m$ with probability $c$. Note that the third transition is done under the assumption of a uniform prior on $m$.

To compute the denominator, we expand:

$$\begin{aligned}
\sum_{m'=0}^{\infty} \Pr(r \mid c,m') &= \sum_{m'=r}^{\infty} \binom{m'}{r} c^r (1-c)^{m'-r} \\
&= c^r \sum_{k=0}^{\infty} \binom{r+k}{r} (1-c)^k = c^r \sum_{k=0}^{\infty} \frac{(r+k)!}{k!r!} (1-c)^k \\
&= c^r \sum_{k=0}^{\infty} \frac{(-1)^k}{k!} (r+1)(r+2)\ldots(r+1+(k-1))(c-1)^k \\
&= c^r \sum_{k=0}^{\infty} \frac{1}{k!} (-(r+1))(-(r+2))\ldots(-(r+k))1^{r+k+1}(c-1)^k \\
&\text{by Taylor series centered in } 1 = c^r \frac{1}{c^{r+1}} = \frac{1}{c}
\end{aligned}$$

Thus,

$$\begin{aligned}
\Pr(\hat{\Psi} \mid \Psi,r,c) &= \sum_{m=r}^{\infty} \Pr(\hat{\Psi} \mid \Psi,r,c,m) \cdot \Pr(m \mid r,c) \\
&= \sum_{m=r}^{\infty} \frac{\binom{m\Psi}{r\hat{\Psi}} \binom{m(1-\Psi)}{r(1-\hat{\Psi})} }{\binom{m}{r}} \frac{\binom{m}{r} c^r (1-c)^{m-r}}{\frac{1}{c}} \\
&\approx \sum_{m=r}^{10r/c} \binom{m\Psi}{r\hat{\Psi}} \binom{m(1-\Psi)}{r(1-\hat{\Psi})} \cdot c^{r+1} (1-c)^{m-r}
\end{aligned}$$

In the last equation, we estimate the sum going only up to a large value of $m$, since its posterior probability diminishes. In expectation $m \approx r/c$. We use ten times this value as the maximum.

We can use this equation to estimate the expected proportion of binary $\hat{\Psi}$ observations ($\hat{\Psi}=0$ or 1) that is expected when we observe only $r$ junctions from a splicing event with a given true rate $\Psi$ (*Figure 3—figure supplement 1a*). We can also estimate the chance to have an empirical $\hat{\Psi}$ that is at least within a certain delta (in absolute terms) from the real $\Psi$. Namely, we can estimate $\Pr(\mid \hat{\Psi} - \Psi \mid < \delta \mid \Psi, r, c)$ (*Figure 3—figure supplement 1d*).

In another calculation of interest, one can ask how many mRNA molecules should a gene have in a cell in order to correctly estimate the splicing rate, under a limited capture efficiency $c$. To estimate it, we denote by $d$ a binary variable indicating that the gene has been detected (i.e. $r>0$) and marginalize $r$ in the following way:

$$\Pr(\hat{\Psi} \mid \Psi,c,m,d) = \frac{\Pr(\hat{\Psi},d \mid \Psi,c,m)}{\Pr(d \mid c,m)}$$

Where

$$\begin{aligned}
\Pr(\hat{\Psi}, d \mid \Psi, c, m) &= \sum_{r=1}^{m} \Pr(\hat{\Psi}, r \mid \Psi, c, m) \\
&= \sum_{r=1}^{m} \Pr(\hat{\Psi} \mid \Psi, c, m, r) \cdot \Pr(r \mid c, m) \\
&= \sum_{r=1}^{m} \frac{\binom{m\Psi}{r\hat{\Psi}}\binom{m(1-\Psi)}{r(1-\hat{\Psi})}}{\binom{m}{r}} \cdot \binom{m}{r} c^r (1-c)^{m-r} \\
&= \sum_{r=1}^{m} \binom{m\Psi}{r\hat{\Psi}}\binom{m(1-\Psi)}{r(1-\hat{\Psi})} c^r (1-c)^{m-r}
\end{aligned}$$

and

$$\Pr(d \mid c, m) = 1 - (1-c)^m$$

We use this in *Figure 3—figure supplement 1b* to plot the chances to see only one isoform (binary $\hat{\Psi}$) for a fixed $\Psi$ (set to 0.5) as a function of the number of molecules present in the cell ($m$).

## Probabilistic simulator of splicing in single cell data

1) Biological process. We simulate the expression of 1500 hypothetical genes in 300 cells using Sym-Sim, an in silico simulator of gene expression in single cells (*Zhang et al., 2019*); the expression of gene $g$ in cell $i$ is annotated as $X_i$. We simulate one cassette exon $j$ for each gene $g$. For each cassette, exon $j$ in cell $i$, we simulate an underlying splicing distribution $\Psi_{ij}$ as a Beta distribution with exon-specific parameters $\alpha_j$ and $\beta_j$. The splicing of $j$ in $i$ is simulated as a binomial sampling from $X_{ig}$ with probability $\Psi_{ij}$. 2) Technical process. We simulate the capture, fragmentation and sequencing of each transcript using a modified version of SymSim's True2ObservedCounts function and a random vector of transcript lengths. Finally, we subsample the obtained reads based on the transcript length in order to simulate the coverage of informative splice junctions.

We simulate the splicing of cassette exons in a set of genes $G$ expressed in a population of cells $N$. For each gene $g \in G$, we simulate the splicing of one cassette exon $j$. The inclusion of $j$ forms the isoform $j_A$, while the exclusion of the exons forms the isoform $j_B$. The production of mRNA molecules from $j_A$ in a single cell $i \in N$ is determined by the total expression of $g$, and by the action of the splicing machinery of $i$.

### Biological process
### Splicing from pre-mRNA transcripts in individual cells

For each alternatively spliced gene $g$ in a cell $i$, we simulate the expression of $g$ and the splicing of its cassette exon.

| | |
|---|---|
| $X_{ig}$ | expression of gene $g$ in $n$. |
| $\alpha_j, \beta_j$ | splicing rate distribution parameters. |
| $\Psi_{ij}$ | $\sim \mathrm{Beta}(\alpha_j, \beta_j)$ |
| $X_{A_{ij}}$ | $\sim \mathrm{Binomial}(X_{ig}, \Psi_{ij})$ |
| $X_{B_{ij}}$ | $= X_{ig} - X_{A_{ij}}$ |
| $\Psi_{T_{ij}}$ | $= X_{A_{ij}}/X_{ig}$ |

$X_{ig}$ represents the total number of pre-mRNA transcribed from each gene across all cells. We simulate the total counts using SymSim, an in-silico approach for simulation of single cell gene expression by accounting for the biological sources of variation (*Zhang et al., 2019*). $\Psi_{ij}$ referred to as the underlying splicing rate, is the probability of splicing in the cassette exon $j$ of gene $g$ in cell $i$. Notice that in this simulation, each gene $g$ only has one cassette exon $j$. In a biological context, $\Psi_{ij}$ would be determined by intrinsic attributes of the cassette exon inherent of $g$ (e.g. sequence, secondary structure, binding sites), and by the profile of splicing factors expressed in $n$. $X_{A_{ij}}$ and $X_{B_{ij}}$ are respectively the counts of mRNA molecules from isoforms $g_A$ and $g_B$ in $i$. Notice that $X_{A_{ij}}$ is a random binomial sample from the total number of expressed pre-mRNA molecules of $g$ in $i$ with a probability $\Psi_{ij}$, as it has been modeled before (*Waks et al., 2011*; *Xiong et al., 2011*; *Faigenbloom et al., 2015*;

*Shen et al., 2014*). $\Psi_{T_{ij}}$ is the true isoform ratio that include cassette exon $j$ in cell $i$, obtained as the proportion of molecules of gene $g$ that include the cassette exon $j$.

The distribution of $\Psi_{ij}$ across all cells $i \in N$ is modeled as a Beta distribution, which has been used in previous studies of single cell splicing (*Barash et al., 2010*; *Song et al., 2017*; *Linker et al., 2019*). In this model, the distribution is determined by the parameters $\alpha_j, \beta_j \in (0, \infty)$. The values of these parameter determine the distribution of $\Psi_{ij}$ across all cells as follows:

- Unimodal with intermediate mode if $\alpha_j, \beta_j > 1$.
- Unimodal with mode 1 if $0 < \alpha_j < 1 \leq \beta_j$.
- Unimodal with mode 0 if $0 < \beta_j < 1 \leq \alpha_j$.
- Bimodal with modes 0 and 1 if $0 < \alpha_j, \beta_j, < 1$.
- Uniform if $\alpha_j = \beta_j = 1$.

Notice that $\frac{\alpha_j}{\alpha_j + \beta_j} = \mu(\Psi_{ij})$. By controlling the $\alpha_j, \beta_j$ parameters we can compare the biological underlying distribution of the exon splicing rate with the observed distribution of $\Psi$ inferred from single cell RNA-seq data.

To compare the results of our simulations under the binary-bimodal (alternative isoforms are present in the population, but rarely in the same cell; *Figure 3a,c,d*) and non-binary unimodal (both isoforms regularly appear in the same cell; *Figure 3b,e,f*) models of splicing, we simulated 500 alternatively spliced exons for each model. For the first model, we simulated the underlying splicing distribution of each exon as bimodal Beta distributions. For the second model, we simulated the underlying splicing distributions of each exon with unimodal Beta distributions with intermediate mode. For each cassette exon $j$ in cell $i$:

- Binary-bimodal splicing:

$$\Psi_{B_{ij}} \sim \mathrm{Beta}(\alpha_{B_j}, \beta_{B_j})$$
$$\alpha_{B_j}, \beta_{B_j} \sim \frac{1}{\mathrm{Uniform}(1, 30)}$$

- Non-binary unimodal splicing:

$$\Psi_{U_{ij}} \sim \mathrm{Beta}(\alpha_{U_j}, \beta_{U_j})$$
$$\alpha_{U_j}, \beta_{U_j} \sim \frac{1}{\mathrm{Uniform}(1, 30)}$$

To simulate a realistic scenario in both models, we simulated 500 additional exons that are consistently included, and 500 that are consistently excluded. For the consistently included exons, we sampled the parameters for Unimodal Beta distributions with mode 1 as

$$\alpha_j \sim \mathrm{Uniform}(1, 30); \beta_j \sim \frac{1}{\mathrm{Uniform}(1, 5)}$$

For the consistently excluded exons, we sampled the parameters for Unimodal Beta distributions with mode 0 as

$$\alpha_j \sim \frac{1}{\mathrm{Uniform}(1, 5)}; \beta_j \sim \mathrm{Uniform}(1, 30)$$

## Technical process
### mRNA capture into cDNA

After simulating the production of mRNAs of distinct isoforms in single cells, we simulate the process of capture and sequencing of mRNA molecules from $X_{A_{ij}}$ and $X_{B_{ij}}$.

$c$ expected capture efficiency
$c_{I_{ng}}$ drawn from truncated normal with mean $c$ and variance 0.002
$C_{I_{ng}} \sim \mathrm{Binomial}\,(X_{I_{ng}}, c_{I_{ng}})$

The process of mRNA capture is simulated using SymSim. $C_{I_{ng}}$ is the number of mRNA molecules

of isoform $I \in \{g_A, g_B\}$ for gene $g$ on cell $n$. This number is sampled from the total number of molecules for isoform that are present in the cell. $c$ is a parameter that determines the expected capture efficiency. $c_{I_{ng}}$ is the specific probability of capture of isoform $I$ of gene $g$ in cell $n$, which is drawn from a truncated normal distribution with mean $c$. The variance of the truncated normal distribution is set at 0.002, which is the default variance in SymSim. The distribution is truncated at 0 and at 1.

## RNA sequencing

Sequencing is also simulated using SymSim's approach, which includes a length-dependent PCR amplification bias modeled from experimental data.

$$
\begin{aligned}
l_{g_B} &\quad \text{sampled from SymSim's database} \\
l_{e_A} &\quad \text{sampled from exon database} \\
l_{g_A} &= l_{g_B} + l_{e_A} \\
R_{I_{ng}} &= \text{True2ObservedCounts}(C_{I_{ng}}, l_{g_I})
\end{aligned}
$$

$l_{g_B}$ is the length of the mRNA transcript of isoform $g_B$ of gene $g$, which represents the isoform that skips the alternative exon. $l_{g_A}$ is the length of the mRNA transcript of the isoform $g_A$, which includes the exon. We assigned the lengths to the excluded isoform by drawing without replacement from SymSim's transcript length database. $l_{e_A}$ is the length of the alternative exon included in isoform $g_A$; it is sampled without replacement from a database of skipped exon lengths from the human genome. $R_{I_{ng}}$ is the total number of observed reads from isoform $I$ of gene $g$ in cell $n$ obtained from single cell RNA sequencing. These are obtained using the True2ObservedCounts function from SymSim with the modification previously described.

## Splice junction coverage and observed Ψ calculation

We also simulate the down-sampling from observing only reads that overlap the splice junctions that are informative about the splicing of the cassette exon.

$$
\begin{aligned}
l_r &= \text{read length (constant)} \\
j_{A_g} &= \frac{4(l_r - 1)}{l_{A_g}} \\
j_{B_g} &= \frac{2(l_r - 1)}{l_{B_g}} \\
SJ_{I_{ng}} &\sim \text{Binomial}(R_{I_{ng}}, j_{I_g})
\end{aligned}
$$

$l_r$ corresponds to the constant read length from the sequencing process. $SJ_{I_{ng}}$ is the number of reads that cover informative splice junctions for isoform $I \in \{g_A, g_B\}$ for gene $g$ in cell $n$, which are sampled from the total number of reads covering the isoform. In order to account for variation in read density along transcripts, we sample the splice junction reads of each event from a binomial distribution with probabilities $j_{A_g}$ and $j_{B_g}$, which are respectively the probabilities of a given read to cover the splice junctions informative with isoform $g_A$ and $g_B$. We derive these probabilities from the length of the simulated reads, and the length of the transcript. Each read can be mapped to $2(l_r - 1)$ positions in the transcript that overlap one splice junction. Thus, the probability of covering one given splice junction is defined as the number of possible positions in the transcript that are informative for the splice junction, divided by the length of the transcript. $j_{A_g}$ is the probability to map to any of the two splice junctions that are informative for isoform $g_A$. $j_{B_g}$ is the probability to map to one single splice junction, since there is only one junction informative for isoform $g_B$. Coverage depth variance might sometimes exceed the variance of the binomial distribution, such that the informative reads for one isoform might be much more likely to be sequenced than the informative reads of the other isoform in a given cell. In these situations, the technical variance of $\hat{\Psi}$ and the number of spurious binary observations could be higher than modeled in these simulations.

Finally, the observed Ψ is calculated as:

$$
\hat{\Psi}_{ij} = \frac{SJ_{A_{ij}}}{SJ_{A_{ij}} + SJ_{B_{ij}}}
$$

## Simulator variants for studying sources of variation

### Gene expression and underlying $\Psi$

We tested the effect of the interplay between gene expression and the ratio of isoforms that contain the cassette exon on the observed distribution of $\Psi$. For this test, we simulated a population of 300 single cells with 500 genes, indexed 1 to 500. For every cell $i$, the expression of gene $g$ is fixed as $X_{ig} = g$, where $g \in \{1, 2, ..., 500\}$. That is, every gene had a different level of expression, and the expression of every individual gene was constant across all cells. For each simulation, we fixed the underlying splicing rate of all cassette exons across all cells. That is, for each cassette exon $j$ of gene $g$, in every cell, we set $\Psi_{ij} = $ constant. We ran the simulator with different underlying splicing rates, with $\Psi_{ij} \in \{0.01, 0.02, ..., 0.5\}$. For every simulation, we used an average capture efficiency $c = 0.1$. We ran 30 simulations for every fixed $\Psi_{ij}$ value. For every fixed $\Psi_{ij}$ and for every fixed expression level $g$, we took the average proportion of cells with binary values ($\hat{\Psi} = 0$ or 1) for the observed $\hat{\Psi}$. That is, we reported:

$$\frac{1}{1500} \sum_{\text{sim}=1}^{30} \sum_{i=1}^{300} \mathbb{I}(\hat{\Psi}_{ij} = 1) + \mathbb{I}(\hat{\Psi}_{ij} = 0)$$

### Gene expression and capture efficiency

We tested the effect of capture efficiency in $\Psi$ observations. To minimize the effect of the underlying $\Psi$ in the simulations, in this analysis we fixed the true splicing rate of all exons to $\Psi_{Tng} = 0.5$ (we achieved this by setting $X_{Aij} = X_{Bij} = \frac{1}{2}X_{ig}$). We ran simulations for each possible value for the average capture efficiency in $c \in \{0.01, 0.011, ..., 0.1\}$. For each tested average capture efficiency rate, we ranked the alternative splicing events by the number of reads that cover the informative splice junctions. For each alternative event, we observed the proportion of cells that present only one type of isoform (either including the cassette exon or excluding it, but not both).

## Acknowledgements

We thank Don Rio for discussions that inspired this analysis, Chenling Xu and Amanda Mok for insightful suggestions, and Nicholas Ingolia for critical comments on the manuscript. C.F.B. was supported by the UC MEXUS-CONACYT doctoral fellowship. N.Y. was supported by the Chan Zuckerberg Biohub.

## Additional information

### Funding

| Funder | Grant reference number | Author |
|---|---|---|
| UC MEXUS-Conacyt | Doctoral Fellowship | Carlos F Buen Abad Najar |
| Chan Zuckerberg Biohub | | Nir Yosef |

The funders had no role in study design, data collection and interpretation, or the decision to submit the work for publication.

### Author contributions

Carlos F Buen Abad Najar, Conceptualization, Software, Formal analysis, Investigation, Methodology, Writing - original draft; Nir Yosef, Liana F Lareau, Conceptualization, Formal analysis, Supervision, Methodology, Writing - original draft

### Author ORCIDs

Nir Yosef ⓘ https://orcid.org/0000-0001-9004-1225
Liana F Lareau ⓘ https://orcid.org/0000-0003-3223-3426

### Decision letter and Author response

Decision letter https://doi.org/10.7554/eLife.54603.sa1

Author response https://doi.org/10.7554/eLife.54603.sa2

## Additional files

### Supplementary files

- Transparent reporting form

### Data availability

All sequencing data reanalyzed in this study were acquired from GEO.

The following previously published datasets were used:

| Author(s) | Year | Dataset title | Dataset URL | Database and Identifier |
|---|---|---|---|---|
| Chen G, Schell JP, Benitez JA, Petropoulos S, Yilmaz M, Reinius B, Alekseenko Z, Shi L, Hedlund E, Lanner F, Sandberg R, Deng Q | 2016 | Single-cell analysis of allelic gene expression in pluripotency, differentiation and X-chromosome inactivation | https://www.ncbi.nlm.nih.gov/geo/query/acc.cgi?acc=GSE74155 | NCBI Gene Expression Omnibus, GSE74155 |
| Lescroart F, Wang X, Lin X, Swedlund B, Gargouri S, Sànchez-Dànes A, Moignard V, Dubois C, Paulissen C, Kinston S, Göttgens B, Blanpain C | 2018 | Defining the early steps of cardiovascularlineage segregation by single cell RNA-seq | https://www.ncbi.nlm.nih.gov/geo/query/acc.cgi?acc=GSE100471 | NCBI Gene Expression Omnibus, GSE100471 |
| Trapnell C, Cacchiarelli D, Grimsby J, Pokharel P, Li S, Morse M, Mikkelsen T, Rinn J | 2014 | Pseudo-temporal ordering of individual cells reveals regulators of differentiation | https://www.ncbi.nlm.nih.gov/geo/query/acc.cgi?acc=GSE52529 | NCBI Gene Expression Omnibus, GSE52529 |
| Song Y, Botvinnik OB, Lovci MT, Kakaradov B, Liu P, Xu JL, Yeo GW | 2017 | Single-cell alternative splicing analysis with Expedition reveals splicing dynamics during neuron differentiation | https://www.ncbi.nlm.nih.gov/geo/query/acc.cgi?acc=GSE85908 | NCBI Gene Expression Omnibus, GSE85908 |
| Fletcher RB, Das D, Gadye L, Street KN, Baudhuin A, Wagner A, Cole MB, Flores Q, Choi YG, Yosef N, Purdom E, Dudoit S, Risso D, Ngai J | 2017 | Olfactory stem cell differentiation: horizontal basal cell (HBC) lineage | https://www.ncbi.nlm.nih.gov/geo/query/acc.cgi?acc=GSE95601 | NCBI Gene Expression Omnibus, GSE95601 |

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
