## [Decision Letter]

**Acceptance summary:**

The manuscript provides substantial value for the splicing community through demonstrating that previous reports of binary splicing in single cells largely result from technical limitations of single-cell RNA-seq (scRNA-seq). Importantly, the findings provide a path forward for future analysis of alternative splicing regulation in single cells.

**Decision letter after peer review:**

Thank you for submitting your article "Coverage-dependent bias creates the appearance of binary splicing in single cells" for consideration by *eLife*. Your article has been reviewed by two peer reviewers, one of whom is a member of our Board of Reviewing Editors, and the evaluation has been overseen by Patricia Wittkopp as the Senior Editor. The reviewers have opted to remain anonymous.

The reviewers have discussed the reviews with one another and the Reviewing Editor has drafted this decision to help you prepare a revised submission.

Summary:

In this manuscript, Najar et al. demonstrate that previous reports of binary splicing in single cells largely result from technical limitations of single-cell RNA-seq (scRNA-seq). The authors challenge the rigorousness of prior published conclusions related to the occurrence of "bimodal" splicing among isogenic cells based on single-cell RNA sequencing – bimodal referring to a situation in which individual cells express either one or the other of two mRNA isoforms, with few cells expressing both isoforms. While previous studies observed frequent bimodal splicing, this study concludes that bimodal patterns arise almost entirely from technical limitations of single cell RNA-seq library preparation and sequencing, particularly the limited fraction of mRNA molecules that are captured by these approaches. The authors develop a filtering approach that allows to identify exons that can be accurately analyzed for splicing in single cells.

This manuscript has important implications for the analysis of alternative splicing regulation in single cells. The analyses described are thoughtful and carefully done, the Materials and methods are very clearly described, and the data presented are fairly convincing. However, there are a few issues that need to be addressed for a fully satisfactory treatment of this subject.

Essential revisions:

1) The authors analyze five previously published scRNA-seq datasets (Chen, Lescroart, Trapnell, Song and Fletcher). The authors should explain why these specific datasets were chosen, and the analysis should be extended to the dataset described in Shalek et al., 2013), which was the first report of widespread bimodality in splicing. A table should be included listing how many examples of bimodal splicing were reported in each original paper, how many examples pass the authors' filters, and how many of these are bimodal. In the examples shown in Figure 3F and J, bimodal splicing appears to occur between cells in different states but not between cells in the same state. Additional discussion should be included related to the fundamental issue of whether or not the authors' analysis supports the existence of any authentic examples of bimodal splicing in isogenic cells which are in the same cell state.

2) A few other sources of variability that could impact the inference of splicing patterns in individual cells should be considered by the authors:

a) "We assume that in each cell, the expected number of reads covering a splice junction is the same as the number of reads expected to cover each nucleotide." This ignores the fact that alternative splicing junctions may have a lower fraction of mappable, because the exclusion junction shares sequence with each of the inclusion junctions, potentially yielding an increased frequency of multi-mapping (or mis-mapping) of reads deriving from alternative isoforms. How does this phenomenon impact detection of bimodal splicing?

b) "We assume that the distribution of reads is expected to be uniform across the transcript." This is another assumption that needs to be explored, since the density of RNA-seq reads along transcripts is notoriously variable. The authors could model the empirical variability in read density along constitutive portions of transcripts in each dataset and apply this model to address how this variability impacts inference of splicing levels.

c) Mammalian gene expression is intrinsically bursty (e.g., PMID 30602787). The authors should discuss whether their analysis captures this effect or how burstiness might impact splicing detection.

d) The amount of pre-mRNA varies across genes and conditions in scRNA-seq datasets (e.g., PMID 30089906). How might this variability influence the results?

3) The authors' validation of their filtering procedure is underdeveloped. The main approach is based on the intuition that genes with truly bimodal splicing should display a higher degree of coregulation, for which the covariance structure of the data is used as a proxy. In Figure 3I, some of the controls are not discussed in the text. Furthermore, the random control appears to perform nearly as well as the true filter in the Trapnell and Song datasets, while in the Chen dataset it is not clear that the filtering leads to much improvement compared to no filter. If this line of argument is to be pursued, a more rigorous analysis and interpretation of the filter's performance relative to controls (perhaps relating to cell subtypes) is needed in the text. Below are specific questions regarding this analysis:

a) In Figure 3I, the authors observe that "the combined filter recovered more evidence of co-regulation than the simple read-based filter". However, the difference between the combined filter and the random filter is not very pronounced. Is this statistically significant?

b) Could the difference between the combined filter and the read-based filter result in part from the lower number of observations?

c) In addition, can the authors justify why this analysis was performed on only 3/5 datasets?

d) It would be helpful to indicate in the text how many exons pass the filters in each dataset and/or what proportion of exons covered this represents.

---

## [Author Response]

Essential revisions:1) The authors analyze five previously published scRNA-seq datasets (Chen, Lescroart, Trapnell, Song and Fletcher). The authors should explain why these specific datasets were chosen, and the analysis should be extended to the dataset described in Shalek et al., 2013), which was the first report of widespread bimodality in splicing.

We began studying splicing in scRNA-seq as part of a separate project, and most of the datasets included here – Chen, Song, Lescroart, and Fletcher – were chosen because they had full-length scRNA-seq of neural or cardiomyocyte differentiation timecourses that were relevant to that project. We now begin the Results section with this explanation of the initial motivation. When we implemented the Census normalization described by Qiu et al., 2017, we included the Trapnell dataset that was the basis of their analysis. Initially we were hesitant to include the Shalek et al., 2013 dataset due to its very small number of cells. However, we agree that this paper was an important, early contribution to the field and we have included analysis of their data in the updated version of the paper. We report similar observations in this dataset that we found in the others.

A table should be included listing how many examples of bimodal splicing were reported in each original paper, how many examples pass the authors' filters, and how many of these are bimodal. In the examples shown in Figure 3F and J, bimodal splicing appears to occur between cells in different states but not between cells in the same state. Additional discussion should be included related to the fundamental issue of whether or not the authors' analysis supports the existence of any authentic examples of bimodal splicing in isogenic cells which are in the same cell state.

We have substantially changed our presentation to address this important point. Our revision includes a new table showing the proportion of bimodal observations before filtering and its drastic decrease upon selection of good data. We have added substantial analysis to look specifically at bimodal splicing in cells in the same state, first clustering the cells by gene expression to separate cell types.

Of the six datasets analysed for this paper, only Shalek et al., 2013 and Song et al., 2017 made specific claims of bimodal splicing observations. One challenge is in finding a good definition of bimodality; the fit test used in Song et al. was prone to calling splicing bimodal if it did not fit an unrealistically narrow set of parameters for unimodality. For this reason, we had previously avoided making a direct comparison to the individual classifications in their paper. To create our new table, we chose a definition of bimodality based on the distribution of the data towards extreme values, to standardize across datasets.

As shown in the new table and accompanying text in our revised manuscript, we found that the number of exons classified as bimodal dropped from 25-50% of exons to none or only a handful of exons in each cell type among all datasets, after we applied our filters. This new analysis supports our previous conclusion that the vast majority of events classified as bimodal are the result of the distortion caused by low capture efficiency in single cells. We are hesitant to make an absolute claim that binary, bimodal splicing does not exist among homogeneous cells of the same type, but it seems possible that the reported examples all represent technical errors or hidden cell subtypes. In line with this, we found that the only remaining examples of binary, bimodal splicing in the Chen dataset represented sex-specific splicing differences. A detailed version of the analysis presented in our revised manuscript is available in our github:

https://github.com/lareaulab/sc_binary_splicing/blob/master/results/data_analysis/Table_1_and_2_reviewed.ipynb

2) A few other sources of variability that could impact the inference of splicing patterns in individual cells should be considered by the authors:a) "We assume that in each cell, the expected number of reads covering a splice junction is the same as the number of reads expected to cover each nucleotide." This ignores the fact that alternative splicing junctions may have a lower fraction of mappable, because the exclusion junction shares sequence with each of the inclusion junctions, potentially yielding an increased frequency of multi-mapping (or mis-mapping) of reads deriving from alternative isoforms. How does this phenomenon impact detection of bimodal splicing?

This is an important point, and we have updated our analysis to use the expected number of reads covering a splice junction. We calculated this number (for each cell individually) as the number of reads covering all constitutive splice junctions, divided by the estimated number of splice junctions present in the pool of mRNA molecules in the cell. The number of constitutive splice junctions per gene was multiplied by the number of estimated mRNA molecules obtained from the Census normalization.

As the reviewer noted, this new value is slightly lower than the reads per nucleotide that we used in the previous version of the manuscript, but closely correlated, and our new method did not change our results.

b) "We assume that the distribution of reads is expected to be uniform across the transcript." This is another assumption that needs to be explored, since the density of RNA-seq reads along transcripts is notoriously variable. The authors could model the empirical variability in read density along constitutive portions of transcripts in each dataset and apply this model to address how this variability impacts inference of splicing levels.

This was poorly phrased in our text. Our simulation does include variation of read coverage along the transcripts – we sample the reads from a binomial distribution. We derive the probability of this binomial distribution from the average coverage. We have updated the Materials and methods text to make this clear.

c) Mammalian gene expression is intrinsically bursty (e.g., PMID 30602787). The authors should discuss whether their analysis captures this effect or how burstiness might impact splicing detection.

This is an interesting idea. Although our overall conclusion is that very little binary splicing exists biologically, transcriptional bursting could lead to correlated splicing outcomes in a certain time window. We added an analysis of the effect of transcriptional burst parameters (size and frequency of bursts) on the incidence of binary splicing observations (Figure 2—figure supplement 1C, D). We found that genes with large bursts or frequent bursts had fewer binary observations, but this effect was due to their (directly related) higher expression and mRNA abundance. We did not see any effect of burstiness beyond overall expression in a regression analysis.

d) The amount of pre-mRNA varies across genes and conditions in scRNA-seq datasets (e.g., PMID 30089906). How might this variability influence the results?

Our results show that the uncertainty of the observed Ψ (and the likelihood of binary observations) is determined by the number of recovered, informative mRNA molecules. We use the Census approach to estimate the number of recovered mRNA molecules, but this could be distorted slightly by differences in pre-mRNA abundance, because some reads coming from pre-mRNA will map to exons and thus contribute to the TPM and Census counts. These pre-mRNAs will not be informative about the splicing outcome. So, more pre-mRNA would lead us to overestimate the number of recovered, informative mRNAs. However, our filtering approach does account for such situations, because we consider the number of splice junction reads that we actually see, and filter out alternative exons where the number of splice junction reads is lower than we would expect from the estimated number of mRNAs.

3) The authors' validation of their filtering procedure is underdeveloped. The main approach is based on the intuition that genes with truly bimodal splicing should display a higher degree of coregulation, for which the covariance structure of the data is used as a proxy. In Figure 3I, some of the controls are not discussed in the text. Furthermore, the random control appears to perform nearly as well as the true filter in the Trapnell and Song datasets, while in the Chen dataset it is not clear that the filtering leads to much improvement compared to no filter. If this line of argument is to be pursued, a more rigorous analysis and interpretation of the filter's performance relative to controls (perhaps relating to cell subtypes) is needed in the text.

We have replaced our coregulation-based analysis of the filters with a new approach that more carefully examines how the filters remove exons whose technical variance overwhelms biological variance, and we show that the measurements of the remaining exons allow identification of differential splicing between cell types. This was a substantial new effort and we hope it addresses the reviewers’ thoughtful comments. Importantly, we clarify our main point that the interpretation of a certain number of splice junction reads is quite different between different datasets. More detailed responses are below:

Below are specific questions regarding this analysis:a) In Figure 3I, the authors observe that "the combined filter recovered more evidence of co-regulation than the simple read-based filter". However, the difference between the combined filter and the random filter is not very pronounced. Is this statistically significant?

We re-structured the validation approach for our filtering procedure. This new approach focuses on the biological significance of the variability of the selected observations. For this, we tested if the exons selected by the filtering procedure are more likely to show statistically significant Ψ differences between cell types, than the exons that are not selected, using the Kruskal-Wallis analysis of variance test for assessing Ψ changes between cell types. (This test reflects the variance of the measurements; exons with noisy data are just as likely to have true biological changes, but the noisy data are unlikely to produce a significant p-value for these changes.) We also evaluated the filters using the autocorrelation test described in DeTomaso et al., 2019, calculating the p-values with an approach that explicitly lessens the effect of missing observations.

Our text was also changed to clarify the main purpose of this section. The goal of this analysis is to show that the naive use of the number of splice junction reads as the main indicator of the quality of Ψ observations can be misleading. Variation in capture efficiency, amplification and read coverage between datasets can lead to differences in the relationship between captured mRNA molecules and observed junction read coverage. As a result, the observation of a fixed number of splice junction reads can have widely different implications between different datasets. This is best exemplified in the differences between the Chen and Song datasets. The former has a higher median of estimated mRNA molecules captured per cell than the latter, but the latter has a much higher number of reads, possibly due to overamplification, As a result, a requirement of 10 reads per splice junction in the Chen dataset is a strict threshold that captures only the most informative alternative splicing events. In contrast, the same number of reads in the Song dataset is a loose threshold that is met by many exons that we suspect that are observed in a very small number of captured mRNAs per cell.

Therefore, since the analysis in the earlier sections show that the capture of mRNA molecules is one of the most important limiting factors for accurately observing splicing events, we propose to take a data-driven approach to the study of splicing in single cells. Instead of relying only on the number of splice junction reads, biologists should also consider other factors such as the number of mRNA molecules captured in the sequencing experiment, and the read coverage depth.

We show that a group of exons selected based on the number of estimated captured mRNA molecules consistently recovers exons with a high incidence of significant differential splicing. However, selecting exons based on a flat threshold of splice junction leads to widely different results between datasets. These results open new questions and possibilities for statistical analysis of single cell splicing based on a better understanding of the data, its biases and its limitations.

b) Could the difference between the combined filter and the read-based filter result in part from the lower number of observations?

This should not be a strong concern for our new validation approach. We did not observe a strong correlation between the number of observations and the significance (p-value) of differential splicing with the Kruskal-Wallis test. Moreover, our autocorrelation test implementation specifically accounts for differences in the number of observations for the calculation of p-values. The results from both tests are consistent.

c) In addition, can the authors justify why this analysis was performed on only 3/5 datasets?

The Lescroart et al. dataset was not considered in this analysis because the two cell types in this study are very similar. Only three exons had a splicing change higher than 0.1. The Fletcher et al. dataset, with very low mRNA recovery, was not included in this analysis because only three exons pass the 10 mRNA filter. Finally, the Shalek dataset (which we added in this revision) is not included due to its low number of cells (13 that are kept after the Census normalization). These reasons are briefly addressed in the updated version of the manuscript.

d) It would be helpful to indicate in the text how many exons pass the filters in each dataset and/or what proportion of exons covered this represents.

We added a new table showing this (Table 1).